# Bounding errors of Expectation-Propagation

**Guillaume Dehaene**
University of Geneva
guillaume.dehaene@gmail.com

**Simon Barthelmé**
CNRS, Gipsa-lab
simon.barthelme@gipsa-lab.fr

## Abstract

Expectation Propagation is a very popular algorithm for variational inference, but comes with few theoretical guarantees. In this article, we prove that the approximation errors made by EP can be bounded. Our bounds have an asymptotic interpretation in the number $n$ of datapoints, which allows us to study EP's convergence with respect to the true posterior. In particular, we show that EP converges at a rate of $\mathcal{O}(n^{-2})$ for the mean, up to an order of magnitude faster than the traditional Gaussian approximation at the mode. We also give similar asymptotic expansions for moments of order 2 to 4, as well as excess Kullback-Leibler cost (defined as the additional KL cost incurred by using EP rather than the ideal Gaussian approximation). All these expansions highlight the superior convergence properties of EP. Our approach for deriving those results is likely applicable to many similar approximate inference methods. In addition, we introduce bounds on the moments of log-concave distributions that may be of independent interest.

## Introduction

Expectation Propagation (EP, 1) is an efficient approximate inference algorithm that is known to give good approximations, to the point of being almost exact in certain applications [2, 3]. It is surprising that, while the method is empirically very successful, there are few theoretical guarantees on its behavior. Indeed, most work on EP has focused on efficiently implementing the method in various settings. Theoretical work on EP mostly represents new justifications of the method which, while they offer intuitive insight, do not give mathematical proofs that the method behaves as expected. One recent breakthrough is due to Dehaene and Barthelmé [4] who prove that, in the large data-limit, the EP iteration behaves like a Newton search and its approximation is asymptotically exact. However, it remains unclear how good we can expect the approximation to be when we have only finite data. In this article, we offer a characterization of the quality of the EP approximation in terms of the worst-case distance between the true and approximate mean and variance.

When approximating a probability distribution $p(x)$ that is, for some reason, close to being Gaussian, a natural approximation to use is the Gaussian with mean equal to the mode (or argmax) of $p(x)$ and with variance the inverse log-Hessian at the mode. We call it the Canonical Gaussian Approximation (CGA), and its use is usually justified by appealing to the Bernstein-von Mises theorem, which shows that, in the limit of a large amount of independent observations, posterior distributions tend towards their CGA. This powerful justification, and the ease with which the CGA is computed (finding the mode can be done using Newton methods) makes it a good reference point for any method like EP which aims to offer a better Gaussian approximation at a higher computational cost. In section 1, we introduce the CGA and the EP approximation. In section 2, we give our theoretical results bounding the quality of EP approximations.

# 1 Background

In this section, we present the CGA and give a short introduction to the EP algorithm. In-depth descriptions of EP can be found in Minka [5], Seeger [6], Bishop [7], Raymond et al. [8].

## 1.1 The Canonical Gaussian Approximation

What we call here the CGA is perhaps the most common approximate inference method in the machine learning cookbook. It is often called the "Laplace approximation", but this is a misnomer: the Laplace approximation refers to approximating the integral $\int p$ from the integral of the CGA. The reason the CGA is so often used is its compelling simplicity: given a target distribution $p(x) = \exp\left(-\phi\left(x\right)\right)$, we find the mode $x^\star$ and compute the second derivatives of $\phi$ at $x^\star$:

$$
\begin{aligned}
x^\star &= \text{argmin}\phi(x) \\
\beta^\star &= \phi''\left(x^\star\right)
\end{aligned}
$$

to form a Gaussian approximation $q(x) = \mathcal{N}\left(x|x^\star, \frac{1}{\beta^\star}\right) \approx p(x)$. The CGA is effectively just a second-order Taylor expansion, and its use is justified by the Bernstein-von Mises theorem [9], which essentially says that the CGA becomes exact in the large-data (large-$n$) asymptotic limit. Roughly, if $p_n(x) \propto \prod_{i=1}^n p\left(y_i|x\right) p_0\left(x\right)$, where $y_1 \ldots y_n$ represent independent datapoints, then $\lim_{n \to \infty} p_n\left(x\right) = \mathcal{N}\left(x|x_n^\star, \frac{1}{\beta_n^\star}\right)$ in total variation.

## 1.2 CGA vs Gaussian EP

Gaussian EP, as its name indicates, provides an alternative way of computing a Gaussian approximation to a target distribution. There is broad overlap between the problems where EP can be applied and the problems where the CGA can be used, with EP coming at a higher cost. Our contribution is to show formally that the higher computational cost for EP may well be worth bearing, as EP approximations can outperform CGAs by an order of magnitude. To be specific, we focus on the *moment estimates* (mean and covariance) computed by EP and CGA, and derive bounds on their distance to the true mean and variance of the target distribution. Our bounds have an asymptotic interpretation, and under that interpretation we show for example that the mean returned by EP is within an order of $\mathcal{O}\left(n^{-2}\right)$ of the true mean, where $n$ is the number of datapoints. For the CGA, which uses the mode as an estimate of the mean, we exhibit a $\mathcal{O}\left(n^{-1}\right)$ upper bound, and we compute the error term responsible for this $\mathcal{O}\left(n^{-1}\right)$ behavior. This enables us to show that, in the situations in which this error is indeed $\mathcal{O}\left(n^{-1}\right)$, EP is better than the CGA.

## 1.3 The EP algorithm

We consider the task of approximating a probability distribution over a random-variable $\mathcal{X} : p(x)$, which we call the *target distribution*. $\mathcal{X}$ can be high-dimensional, but for simplicity, we focus on the one-dimensional case. One important hypothesis that makes EP feasible is that $p(x)$ factorizes into $n$ *simple* factor terms:

$$
p(x) = \prod_i f_i(x)
$$

EP proposes to approximate each $f_i(x)$ (usually referred to as *sites*) by a Gaussian function $q_i(x)$ (referred to as the *site-approximations*). It is convenient to use the parametrization of Gaussians in terms of natural parameters:

$$
q_i\left(x|r_i, \beta_i\right) \propto \exp\left(r_i x - \beta_i \frac{x^2}{2}\right)
$$

which makes some of the further computations easier to understand. Note that EP could also be used with other exponential approximating families. These Gaussian approximations are computed iteratively. Starting from a current approximation $(q_i^t\left(x|r_i^t, \beta_i^t\right))$, we select a site for update with index i. We then:

- Compute the *cavity* distribution $q_{-i}^t(x) \propto \prod_{j \neq 1} q_j^t(x)$. This is very easy in natural parameters:

$$q_{-i}(x) \propto \exp\left( \left( \sum_{j \neq i} r_j^t \right) x - \left( \sum_{j \neq i} \beta_j^t \right) \frac{x^2}{2} \right)$$

- Compute the *hybrid* distribution $h_i^t(x) \propto q_{-i}^t(x) f_i(x)$ and its mean and variance
- Compute the Gaussian which minimizes the Kullback-Leibler divergence to the hybrid, ie the Gaussian with same mean and variance:

$$\mathcal{P}(h_i^t) = \underset{q}{\operatorname{argmin}} \left( KL\left( h_i^t | q \right) \right)$$

- Finally, update the approximation of $f_i$:

$$q_i^{t+1} = \frac{\mathcal{P}(h_i^t)}{q_{-i}^t}$$

where the division is simply computed as a subtraction between natural parameters

We iterate these operations until a fixed point is reached, at which point we return a Gaussian approximation of $p(x) \approx \prod q_i(x)$.

## 1.4 The "EP-approximation"

In this work, we will characterize the quality of an EP approximation of $p(x)$. We define this to be any fixed point of the iteration presented in section 1.3, which could all be returned by the algorithm. It is known that EP will have at least one fixed-point [1], but it is unknown under which conditions the fixed-point is unique. We conjecture that, when all sites are log-concave (one of our hypotheses to control the behavior of EP), it is in fact unique but we can't offer a proof yet. If $p(x)$ isn't log-concave, it is straightforward to construct examples in which EP has multiple fixed-points. These open questions won't matter for our result because we will show that all fixed-points of EP (should there be more than one) produce a good approximation of $p(x)$.

Fixed points of EP have a very interesting characterization. If we note $q_i^*$ the site-approximations at a given fixed-point, $h_i^*$ the corresponding hybrid distributions, and $q^*$ the global approximation of $p(x)$, then the mean and variance of all the hybrids and $q^*$ is the same[1]. As we will show in section 2.2, this leads to a very tight bound on the possible positions of these fixed-points.

## 1.5 Notation

We will use repeatedly the following notation. $p(x) = \prod_i f_i(x)$ is the target distribution we want to approximate. The sites $f_i(x)$ are each approximated by a Gaussian site-approximation $q_i(x)$ yielding an approximation to $p(x) \approx q(x) = \prod_i q_i(x)$. The hybrids $h_i(x)$ interpolate between $q(x)$ and $p(x)$ by replacing one site approximation $q_i(x)$ with the true site $f_i(x)$.

Our results make heavy use of the log-functions of the sites and the target distribution. We note $\phi_i(x) = -\log(f_i(x))$ and $\phi_p(x) = -\log(p(x)) = \sum \phi_i(x)$. We will introduce in section 2 hypotheses on these functions. Parameter $\beta_m$ controls their minimum curvature and parameters $K_d$ control the maximum $d^{th}$ derivative.

We will always consider fixed-points of EP, where the mean and variance under all hybrids and $q(x)$ is identical. We will note these common values: $\mu_{EP}$ and $v_{EP}$. We will also refer to the third and fourth centered moment of the hybrids, denoted by $m_3^i, m_4^i$ and to the fourth moment of $q(x)$ which is simply $3v_{EP}^2$. We will show how all these moments are related to the true moments of the target distribution which we will note $\mu, v$ for the mean and variance, and $m_3^p, m_4^p$ for the third and fourth moment. We also investigate the quality of the CGA: $\mu \approx x^\star$ and $v \approx \left[ \phi_p''(x^\star) \right]^{-1}$ where $x^\star$ is the the mode of $p(x)$.

## 2 Results

In this section, we will give tight bounds on the quality of the EP approximation (ie: of fixed-points of the EP iteration). Our results lean on the properties of log-concave distributions [10]. In section 2.1, we introduce new bounds on the moments of log-concave distributions. The bounds show that those distributions are in a certain sense close to being Gaussian. We then apply these results to study fixed points of EP, where they enable us to compute bounds on the distance between the mean and variance of the true distribution $p(x)$ and of the approximation given by EP, which we do in section 2.2.

Our bounds require us to assume that all sites $f_i(x)$ are $\beta_m$-strongly log-concave with slowly-changing log-function. That is, if we note $\phi_i(x) = -\log(f_i(x))$:

$$\forall i \ \forall x \ \phi_i^{''}(x) \ \geq \ \beta_m > 0 \tag{1}$$

$$\forall i \ \forall d \in [3, 4, 5, 6] \ \left|\phi_i^{(d)}(x)\right| \ \leq \ K_d \tag{2}$$

The target distribution $p(x)$ then inherits those properties from the sites. Noting $\phi_p(x) = -\log(p(x)) = \sum_i \phi_i(x)$, then $\phi_p$ is $n\beta_m$-strongly log-concave and its higher derivatives are bounded:

$$\forall x, \ \phi_p^{''}(x) \ \geq \ n\beta_m \tag{3}$$

$$\forall d \in [3, 4, 5, 6] \ \left|\phi_p^{(d)}(x)\right| \ \leq \ nK_d \tag{4}$$

A natural concern here is whether or not our conditions on the sites are of practical interest. Indeed, strongly-log-concave likelihoods are rare. We picked these strong regularity conditions because they make the proofs relatively tractable (although still technical and long). The proof technique carries over to more complicated, but more realistic, cases. One such interesting generalization consists of the case in which $p(x)$ and all hybrids at the fixed-point are log-concave with slowly changing log-functions (with possibly differing constants). In such a case, while the math becomes more unwieldy, similar bounds as ours can be found, greatly extending the scope of our results. The results we present here should thus be understood as a stepping stone and not as the final word on the quality of the EP approximation: we have focused on providing a rigorous but extensible proof.

### 2.1 Log-concave distributions are strongly constrained

Log-concave distributions have many interesting properties. They are of course unimodal, and the family is closed under both marginalization and multiplication. For our purposes however, the most important property is a result due to Brascamp and Lieb [11], which bounds their even moments. We give here an extension in the case of log-concave distributions with slowly changing log-functions (as quantified by eq. (2)). Our results show that these are close to being Gaussian.

The Brascamp-Lieb inequality states that, if $LC(x) \propto \exp(-\phi(x))$ is $\beta_m$-strongly log-concave (ie: $\phi^{''}(x) \geq \beta_m$), then centered even moments of $LC$ are bounded by the corresponding moments of a Gaussian with variance $\beta_m^{-1}$. If we note these moments $m_{2k}$ and $\mu_{LC} = E_{LC}(x)$ the mean of $LC$:

$$m_{2k} \ = \ E_{LC}\left((x - \mu_{LC})^{2k}\right)$$
$$m_{2k} \ \leq \ (2k-1)!!\beta_m^{-k} \tag{5}$$

where $(2k-1)!!$ is the double factorial: the product of all odd terms from 1 to $2k - 1$. $3!! = 3$, $5!! = 15$, $7!! = 105$, etc. This result can be understood as stating that a log-concave distribution must have a small variance, but doesn't generally need to be close to a Gaussian.

With our hypothesis of slowly changing log-functions, we were able to improve on this result. Our improved results include a bound on *odd* moments, as well as first order expansions of even moments (eqs. (6)-(9)).

Our extension to the Brascamp-Lieb inequality is as follows. If $\phi$ is slowly changing in the sense that some of its higher derivatives are bounded, as per eq. 2, then we can give a bound on $\phi^{'}(\mu_{LC})$

(showing that $\mu_{LC}$ is close to the mode $x^\star$ of $LC$, see eqs. (10) to (13)) and $m_3$ (showing that $LC$ is mostly symmetric):

$$\left|\phi^{'}(\mu_{LC})\right| \leq \frac{K_3}{2\beta_m} \tag{6}$$

$$|m_3| \leq \frac{2K_3}{\beta_m^3} \tag{7}$$

and we can compute the first order expansions of $m_2$ and $m_4$, and bound the errors in terms of $\beta_m$ and the $K$'s :

$$\left|m_2^{-1} - \phi^{''}(\mu_{LC})\right| \leq \frac{K_3^2}{\beta_m^2} + \frac{K_4}{2\beta_m} \tag{8}$$

$$\left|\phi^{''}(\mu_{LC})m_4 - 3m_2\right| \leq \frac{19}{2}\frac{K_3^2}{\beta_m^4} + \frac{5}{2}\frac{K_4}{\beta_m^3} \tag{9}$$

With eq. (8) and (9), we see that $m_2 \approx \left(\phi^{''}(\mu_{LC})\right)^{-1}$ and $m_4 \approx 3\left(\phi^{''}(\mu_{LC})\right)^{-2}$ and, in that sense, that $LC(x)$ is close to the Gaussian with mean $\mu_{LC}$ and inverse-variance $\phi^{''}(\mu_{LC})$.

These expansions could be extended to further orders and similar formulas can be found for the other moments of $LC(x)$: for example, any odd moments can be bounded by $|m_{2k+1}| \leq C_k K_3 \beta_m^{-(k+1)}$ (with $C_k$ some constant) and any even moment can be found to have first-order expansion: $m_{2k} \approx (2k-1)!!\left(\phi^{''}(\mu_{LC})\right)^{-k}$. The proof, as well as more detailed results, can be found in the Supplement.

Note how our result relates to the Bernstein-von Mises theorem, which says that, in the limit of a large amount of observations, a posterior $p(x)$ tends towards its CGA. If we consider the posterior obtained from $n$ likelihood functions that are all log-concave and slowly changing, our results show the slightly different result that the moments of that posterior are *close* to those of a Gaussian with mean $\mu_{LC}$ (instead of $x^\star_{LC}$) and inverse-variance $\phi^{''}(\mu_{LC})$ (instead of $\phi^{''}(x^\star_{LC})$) . This point is critical. While the CGA still ends up capturing the limit behavior of $p$, as $\mu_{LC} \to x^\star$ in the large-data limit (see eq. (13) below), an approximation that would return the Gaussian approximation at $\mu_{LC}$ would be better. This is essentially what EP does, and this is how it improves on the CGA.

## 2.2 Computing bounds on EP approximations

In this section, we consider a given EP fixed-point $q_k^*(x|r_i, \beta_i)$ and the corresponding approximation of $p(x)$: $q^*(x|r = \sum r_i, \beta = \sum \beta_i)$. We will show that the expected value and variance of $q^*$(resp. $\mu_{EP}$ and $v_{EP}$) are close to the true mean and variance of $p$ (resp. $\mu$ and $v$), and also investigate the quality of the CGA ($\mu \approx x^\star$, $v \approx \left[\phi_p^{''}(x^\star)\right]^{-1}$).

Under our assumptions on the sites (eq. (1) and (2)), we are able to derive bounds on the quality of the EP approximation. The proof is quite involved and long, and we will only present it in the Supplement. In the main text, we give a partial version: we detail the first step of the demonstration, which consists of computing a rough bound on the distance between the true mean $\mu$, the EP approximation $\mu_{EP}$ and the mode $x^\star$, and give an outline of the rest of the proof.

Let's show that $\mu$, $\mu_{EP}$ and $x^\star$ are all close to one another. We start from eq. (6) applied to $p(x)$:

$$\left|\phi_p^{'}(\mu)\right| \leq \frac{K_3}{2\beta_m} \tag{10}$$

which tells us that $\phi'_p(\mu) \approx 0$. $\mu$ must thus be close to $x^\star$. Indeed:

$$
\begin{align}
\left| \phi'_p(\mu) \right| &= \left| \phi'_p(\mu) - \phi'_p(x^\star) \right| \tag{11} \\
&= \left| \phi''_p(\xi)(\mu - x^\star) \right| \ \xi \in [\mu, x^\star] \\
&\geq \left| \phi''_p(\xi) \right| |\mu - x^\star| \\
&\geq n\beta_m |\mu - x^\star| \tag{12}
\end{align}
$$

Combining eq. (10) and (12), we finally have:

$$
|\mu - x^\star| \leq n^{-1} \frac{K_3}{2\beta_m^2} \tag{13}
$$

Let's now show that $\mu_{EP}$ is also close to $x^\star$. We proceed similarly, starting from eq. (6) but applied to all hybrids $h_i(x)$:

$$
\forall i \ \left| \phi'_i(\mu_{EP}) + \beta_{-i}\mu_{EP} - r_{-i} \right| \leq n^{-1} \frac{K_3}{2\beta_m} \tag{14}
$$

which is not really equivalent to eq. (10) yet. Recall that $q(x|r,\beta)$ has mean $\mu_{EP}$: we thus have: $r = \beta\mu_{EP}$. Which gives:

$$
\begin{align}
\left( \sum_i \beta_{-i} \right) \mu_{EP} &= ((n-1)\beta)\,\mu_{EP} \\
&= (n-1)r \\
&= \sum_i r_{-i} \tag{15}
\end{align}
$$

If we sum all terms in eq. (14), the $\beta_{-i}\mu_{EP}$ and $r_{-i}$ thus cancel, leaving us with:

$$
\left| \phi'_p(\mu_{EP}) \right| \leq \frac{K_3}{2\beta_m} \tag{16}
$$

which is equivalent to eq. (10) but for $\mu_{EP}$ instead of $\mu$. This shows that $\mu_{EP}$ is, like $\mu$, close to $x^*$:

$$
|\mu_{EP} - x^\star| \leq n^{-1} \frac{K_3}{2\beta_m^2} \tag{17}
$$

At this point, we can show that, since they are both close to $x^\star$ (eq. (13) and (17)), $\mu = \mu_{EP} + \mathcal{O}\left(n^{-1}\right)$, which constitutes the first step of our computation of bounds on the quality of EP.

After computing this, the next step is evaluating the quality of the approximation of the variance, via computing $\left| v^{-1} - v_{EP}^{-1} \right|$ for EP and $\left| v^{-1} - \phi''_p(x^\star) \right|$ for the CGA, from eq. (8). In both cases, we find:

$$
\begin{align}
v^{-1} &= v_{EP}^{-1} + \mathcal{O}\left(1\right) \tag{18} \\
&= \phi''_p(x^\star) + \mathcal{O}\left(1\right) \tag{19}
\end{align}
$$

Since $v^{-1}$ is of order $n$, because of eq. (5) (Brascamp-Lieb upper bound on variance), this is a decent approximation: the relative error is of order $n^{-1}$.

We can find similarly that both EP and CGA do a good job of finding a good approximation of the fourth moment of $p$: $m_4$. For EP this means that the fourth moment of each hybrid and of $q$ are a close match:

$$
\begin{align}
\forall i \ m_4 &\approx m_4^i \approx 3v_{EP}^2 \tag{20} \\
&\approx 3\left( \phi''_p(m) \right)^{-2} \tag{21}
\end{align}
$$

In contrast, the third moment of the hybrids doesn't match at all the third moment of $p$, but their sum does !

$$
m_3 \approx \sum_i m_3^i \tag{22}
$$

Finally, we come back to the approximation of $\mu$ by $\mu_{EP}$. These obey two very similar relationships:

$$\phi_p'(\mu) + \phi_p^{(3)}(\mu)\frac{v}{2} = \mathcal{O}\left(n^{-1}\right) \tag{23}$$

$$\phi_p'(\mu_{EP}) + \phi_p^{(3)}(\mu_{EP})\frac{v_{EP}}{2} = \mathcal{O}\left(n^{-1}\right) \tag{24}$$

Since $v = v_{EP} + \mathcal{O}\left(n^{-2}\right)$ (a slight rephrasing of eq. (18)), we finally have:

$$\mu = \mu_{EP} + \mathcal{O}\left(n^{-2}\right) \tag{25}$$

We summarize the results in the following theorem:

**Theorem 1.** *Characterizing fixed-points of EP*

*Under the assumptions given by eq. (1) and (2) (log-concave sites with slowly changing log), we can bound the quality of the EP approximation and the CGA:*

$$
\begin{aligned}
|\mu - x^*| &\leq n^{-1}\frac{K_3}{2\beta_m^2} \\
|\mu - \mu_{EP}| &\leq B_1(n) = \mathcal{O}\left(n^{-2}\right) \\
\left|v^{-1} - \phi_p''(x^*)\right| &\leq \frac{2K_3^2}{\beta_m^2} + \frac{K_4}{2\beta_m} \\
\left|v^{-1} - v_{EP}^{-1}\right| &\leq B_2(n) = \mathcal{O}\left(1\right)
\end{aligned}
$$

*We give the full expression for the bounds $B_1$ and $B_2$ in the Supplement*

Note that the order of magnitude of the bound on $|\mu - x^\star|$ is the best possible, because it is attained for certain distributions. For example, consider a Gamma distribution with natural parameters $(n\alpha, n\beta)$ whose mean $\frac{\alpha}{\beta}$ is approximated at order $n^{-1}$ by its mode $\frac{\alpha}{\beta} - \frac{1}{n\beta}$. More generally, from eq. (23), we can compute the first order of the error:

$$\mu - m \approx -\frac{\phi_p^{(3)}(\mu)}{\phi_p''(\mu)}\frac{v}{2} \approx -\frac{1}{2}\frac{\phi_p^{(3)}(\mu)}{\left[\phi_p''(\mu)\right]^2} \tag{26}$$

which is the term causing the order $n^{-1}$ error. Whenever this term is significant, it is thus safe to conclude that EP improves on the CGA.

Also note that, since $v^{-1}$ is of order $n$, the relative error for the $v^{-1}$ approximation is of order $n^{-1}$ for both methods. Despite having a convergence rate of the same order, the EP approximation is demonstrably better than the CGA, as we show next. Let us first see why the approximation for $v^{-1}$ is only of order 1 for both methods. The following relationship holds:

$$v^{-1} = \phi_p''(\mu) + \phi_p^{(3)}(\mu)\frac{m_3^p}{2v} + \phi_p^{(4)}(\mu)\frac{m_4^p}{3!v} + \mathcal{O}\left(n^{-1}\right) \tag{27}$$

In this relationship, $\phi_p''(\mu)$ is an order $n$ term while the rest are order 1. If we now compare this to the CGA approximation of $v^{-1}$, we find that it fails at multiple levels. First, it completely ignores the two order 1 terms, and then, because it takes the value of $\phi_p''$ at $x^\star$ which is at a distance of $\mathcal{O}\left(n^{-1}\right)$ from $\mu$, it adds another order 1 error term (since $\phi_p^{(3)} = \mathcal{O}\left(n\right)$). The CGA is thus adding quite a bit of error, even if each component is of order 1.

Meanwhile, $v_{EP}$ obeys a relationship similar to eq. (27):

$$v_{EP}^{-1} = \phi_p''(\mu_{EP}) + \sum_i\left[\phi_i^{(3)}(\mu_{EP})\frac{m_3^i}{2v_{EP}}\right] + \phi_p^{(4)}(\mu_{EP})\frac{3v_{EP}^2}{3!v_{EP}} + \mathcal{O}\left(n^{-1}\right) \tag{28}$$

We can see where the EP approximation produces errors. The $\phi_p''$ term is well approximated: since $|\mu - \mu_{EP}| = \mathcal{O}\left(n^{-2}\right)$, we have $\phi_p''(\mu) = \phi_p''(\mu_{EP}) + \mathcal{O}\left(n^{-1}\right)$. The term involving $m_4$ is also well

approximated, and we can see that the only term that fails is the $m_3$ term. The order 1 error is thus entirely coming from this term, which shows that EP performance suffers more from the skewness of the target distribution than from its kurtosis.

Finally, note that, with our result, we can get some intuitions about the quality of the EP approximation using other metrics. For example, if the most interesting metric is the KL divergence $KL\,(p, q)$, the excess KL divergence from using the EP approximation $q$ instead of the true minimizer $q_{KL}$ (which has the same mean $\mu$ and variance $v$ as $p$) is given by:

$$\Delta KL = \int p \log \frac{q_{KL}}{q} \quad = \quad \int p(x) \left( -\frac{(x-\mu)^2}{2v} + \frac{(x-\mu_{EP})^2}{2v_{EP}} - \frac{1}{2} \log \left( \frac{v}{v_{EP}} \right) \right) \quad (29)$$

$$= \quad \frac{1}{2} \left[ \frac{v}{v_{EP}} - 1 - \log \left( \frac{v}{v_{EP}} \right) \right] + \frac{(\mu-\mu_{EP})^2}{2v_{EP}} \quad (30)$$

$$\approx \quad \frac{1}{4} \left( \frac{v-v_{EP}}{v_{EP}} \right)^2 + \frac{(\mu-\mu_{EP})^2}{2v_{EP}} \quad (31)$$

which we recognize as $KL\,(q_{KL}, q)$. A similar formula gives the excess KL divergence from using the CGA instead of $q_{KL}$. For both methods, the variance term is of order $n^{-2}$ (though it should be smaller for EP), but the mean term is of order $n^{-3}$ for EP while it is of order $n^{-1}$ for the CGA. Once again, EP is found to be the better approximation.

Finally, note that our bounds are quite pessimistic: the true value might be a much better fit than we have predicted here.

A first cause is the bounding of the derivatives of $\log(p)$ (eqs. (3),(4)): while those bounds are correct, they might prove to be very pessimistic. For example, if the contributions from the sites to the higher-derivatives cancel each other out, a much lower bound than $nK_d$ might apply. Similarly, there might be another lower bound on the curvature much higher than $n\beta_m$.

Another cause is the bounding of the variance from the curvature. While applying Brascamp-Lieb requires the distribution to have high log-curvature everywhere, a distribution with high-curvature close to the mode and low-curvature in the tails still has very low variance: in such a case, the Brascamp-Lieb bound is very pessimistic.

In order to improve on our bounds, we will thus need to use tighter bounds on the log-derivatives of the hybrids and of the target distribution, but we will also need an extension of the Brascamp-Lieb result that can deal with those cases where a distribution is strongly log-concave around its mode but, in the tails, the log-curvature is much lower.

## 3 Conclusion

EP has been used for now quite some time without any theoretical concrete guarantees on its performance. In this work, we provide explicit performance bounds and show that EP is superior to the CGA, in the sense of giving provably better approximations of the mean and variance. There are now theoretical arguments for substituting EP to the CGA in a number of practical problems where the gain in precision is worth the increased computational cost. This work tackled the first steps in proving that EP offers an appropriate approximation. Continuing in its tracks will most likely lead to more general and less pessimistic bounds, but it remains an open question how to quantify the quality of the approximation using other distance measures. For example, it would be highly useful for machine learning if one could show bounds on prediction error when using EP. We believe that our approach should extend to more general performance measures and plan to investigate this further in the future.

## Footnotes

[1] For non-Gaussian approximations, the expected values of all sufficient statistics of the exponential family are equal.

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
