[Supplementary Material]

# Supplementary information of "Bounding errors of Expectation-Propagation"

## A  Improving on the Brascamp-Lieb bound

In this section, we detail our mathematical results concerning the extension of the Brascamp-Lieb bound.

We will note $LC(x) = \exp\left(-\phi(x)\right)$ a log-concave distribution. We assume that $\phi$ is strongly convex, and slowly changing, ie:

$$\forall x \; \phi''(x) \;\geq\; \beta_m \tag{32}$$

$$\forall d \in [3,4,5,6] \; \left|\phi^{(d)}(x)\right| \;\leq\; K_d \tag{33}$$

### A.1  The original Brascamp-Lieb theorem

Let $\mu_{LC} = E_{LC}(x)$ be the expected value of $LC$. The original Brascamp-Lieb result [1976] concerns bounding fractional centered moments of $LC$ by the corresponding fractional moments of a Gaussian of variance $\beta_m^{-1}$, centered at $\mu_{LC}$. Noting $g(x) = \mathcal{N}\left(x|\mu_{LC}, \beta_m^{-1}\right)$ that Gaussian, we have:

$$\forall \alpha \geq 1 \; E_{LC}\left(|x - \mu_{LC}|^\alpha\right) \leq E_g\left(|x - \mu_{LC}|^\alpha\right) \tag{34}$$

However, we are not interested in their full result, but only in a restricted version of it which only concerns even moments. This version simply reads:

$$\forall k \in \mathbb{N} \; m_{2k} = E_{LC}\left(|x - \mu_{LC}|^{2k}\right) \;\leq\; (2k-1) \, m_{2k-2} \beta_m^{-1} \tag{35}$$

$$m_{2k} \;\leq\; (2k-1)!! \beta_m^{-k} \tag{36}$$

where $(2k-1)!!$ is the double-factorial: the product of all odd terms between 1 and $2k-1$. Eq. (35) might be a new result. Note that equality only occurs when $f(x) = 1$ and $LC$ is Gaussian. Note also that the bounds on the higher derivatives of $\phi$ are not needed for this result, but only for our extension.

We offer here a proof of eq. (35) (from which eq. (36) is a trivial consequence), which is slightly different from Brascamp & Lieb's original proof. We believe this proof to be original, though it is still quite similar to the original proof.

*Proof.* Let's decompose $LC(x)$ into two parts:

- $g(x) = \mathcal{N}\left(x|\mu_{LC}, \beta_m^{-1}\right)$ the bounding Gaussian with same mean as $LC$

- $f(x) = \frac{LC(x)}{g(x)}$ the remainder

$f$ is easily shown to be log-concave, which means that it is unimodal. We will note $x^\star$ the mode of $f$. $f$ is increasing on $]-\infty, x^\star]$ and decreasing on $[x^\star, \infty[$. We thus know the sign of $f'(x)$:

$$\text{sign}\left(f'(x)\right) = \text{sign}\left(x^\star - x\right) \tag{37}$$

Consider the integral: $\int_{-\infty}^{+\infty} g(x)f'(x)dx$. By integration by parts (or by Stein's lemma), we have:

$$\int_{-\infty}^{+\infty} g(x)f'(x)dx = \int_{-\infty}^{+\infty} g(x)f(x)\beta_m(x-\mu)dx$$

$$= \beta_m(\mu - \mu)$$

$$\int_{-\infty}^{+\infty} g(x)f'(x)dx = 0 \tag{38}$$

We now split the integral at $\mu_{LC}$ and $x^\star$, assuming without loss of generality that $x^\star \leq \mu_{LC}$:

$$\int_{-\infty}^{x^\star} gf' + \int_{x^\star}^{\mu_{LC}} gf' + \int_{\mu_{LC}}^{\infty} gf' = 0$$

$$\int_{-\infty}^{x^\star} gf' + \int_{x^\star}^{\mu_{LC}} gf' = -\int_{\mu_{LC}}^{\infty} gf'$$

$$\geq 0 \qquad (39)$$

Now consider a statistic $S_k(x) = (x - \mu_{LC})^{2k-1}$. Again using integration by parts, we have the following equality:

$$\int g(x) f'(x) S_k(x) dx = \int g(x) f(x) \left( \beta_m S_k(x)(x - \mu) - S_k'(x) \right) dx$$

$$\int g(x) f'(x) (x - \mu_{LC})^{2k-1} dx = \int LC(x) \left( \beta_m (x - \mu_{LC})^{2k} - (2k-1)(x - \mu_{LC})^{2k-2} \right)$$

$$= \beta_m m_{2k} - (2k-1) m_{2k-2} \qquad (40)$$

At this point, we only need to prove that $\int gf' S_k \leq 0$ to finish our proof, from eq. (40). We will actually prove a slightly stronger result: that even if we cut the integral at $\mu_{LC}$, both halves are still negative:

$$\int_{-\infty}^{\mu_{LC}} gf' S_k \leq 0 \qquad (41)$$

$$\int_{\mu_{LC}}^{\infty} gf' S_k \leq 0 \qquad (42)$$

Eq. (42) is trivial. $g$ is positive everywhere, while $S_k(x) \geq 0$ and $f'(x) \leq 0$ for $x \geq \mu_{LC}$.

Eq. (41) is slightly harder. From eq. (39), $\int_{-\infty}^{x^\star} gf' + \int_{x^\star}^{\mu_{LC}} gf' \geq 0$, where the first term is positive, and the second negative. When we multiply the integrand by the decreasing positive function $-S_k(x) = -(x - \mu_{LC})^{2k-1}$, the order in the terms is preserved. To say it in equations:

$$\int_{-\infty}^{x^\star} gf'(-S_k) \geq \left( -(x^\star - \mu_{LC})^{2k-1} \right) \int_{-\infty}^{x^\star} gf'$$

$$\geq \left( -(x^\star - \mu_{LC})^{2k-1} \right) \left( -\int_{x^\star}^{\mu_{LC}} gf' \right)$$

$$\geq \int_{x^\star}^{\mu_{LC}} gf' S_k \qquad (43)$$

from which we finally find eq. (41), which concludes our proof. Note that there is the equality $\int gf' S_k = 0$ IFF $f'(x) = 0$, justifying our earlier comment about $m_{2k} = (2k-1)\beta_m^{-1} m_{2k-2}$ IFF $LC(x) = g(x)$. $\qquad \square$

## A.2  Extending the Brascamp-Lieb theorem

The original Brascamp-Lieb result tells us that the spread of $LC(x)$ (as measured by its even moments) can't be too important, but it doesn't tell us whether such distributions are close to being Gaussian, which is what EP requires. By constraining the higher derivatives of $\phi(x)$, we are able to constrain how far $LC$ is from a Gaussian distribution. This is the essence of our extension of the Brascamp-Lieb theorem. We derived the following:

**Theorem 2.** *Extension of the Brascamp-Lieb theorem*

*With LC a strongly log-concave distribution with slowly changing log-function (eqs. (32), (33)), we have the following inequalities:*

$$\left|\phi^{'}\left(\mu_{LC}\right)\right| \leq \frac{K_3}{2\beta_m} \tag{44}$$

$$\left|\frac{m_3}{m_2}\right| \leq 2\frac{K_3}{\beta_m^2} \tag{45}$$

$$\left|\frac{m_5}{m_2}\right| \leq \frac{17K_3}{\beta_m^3} \tag{46}$$

*which generalizes to:*

$$\left|\frac{m_{2k+1}}{m_2}\right| \leq C_k \frac{K_3}{\beta_m^{k+1}} \tag{47}$$

*The following first order expansions of $m_2$, $m_3$ and $m_4$:*

$$\left|m_2^{-1} - \phi_2^{''}(\mu_{LC})\right| \leq \frac{K_3^2}{\beta_m^2} + \frac{K_4}{2\beta_m} \tag{48}$$

$$\left|\phi^{''}(\mu_{LC})m_2 - 1\right| \leq \frac{K_3^2}{\beta_m^3} + \frac{K_4}{2\beta_m^2} \tag{49}$$

$$\left|\phi^{''}(\mu_{LC})m_3 + \left(\phi^{'}(\mu_{LC})m_2 + \frac{\phi^{(3)}(\mu_{LC})}{2}m_4\right)\right| \leq \frac{17}{6}\frac{K_3K_4}{\beta_m^4} + \frac{5}{8}\frac{K_5}{\beta_m^3} \tag{50}$$

$$\left|\phi^{''}(\mu_{LC})m_4 - 3m_2\right| \leq \frac{19}{2}\frac{K_3^2}{\beta_m^4} + \frac{5}{2}\frac{K_4}{\beta_m^3} \tag{51}$$

*which generalizes to:*

$$m_{2k+2} \approx \frac{(2k+1)}{\phi^{''}(\mu_{LC})}m_{2k} \tag{52}$$

$$\approx (2k+1)!!\left[\phi^{''}(\mu_{LC})\right]^{-(k+1)} \tag{53}$$

*And the following higher order relationships:*

$$\left|\phi^{'}(\mu_{LC}) + \frac{\phi^{(3)}(\mu_{LC})}{2}m_2\right| \leq \frac{K_3K_4}{3\beta_m^3} + \frac{K_5}{8\beta_m^2} \tag{54}$$

$$\left|m_2^{-1} - \phi^{''}(\mu_{LC}) - \frac{\phi^{(3)}(\mu_{LC})}{2}\frac{m_3}{m_2} - \frac{\phi^{(4)}(\mu_{LC})}{3!}\frac{m_4}{m_2}\right| \leq \frac{17}{24}\frac{K_3K_5}{\beta_m^3} + \frac{K_6}{8\beta_m^2} \tag{55}$$

Note that we refer to eq. (48), (50) and (51) as first order expansions because you can read them as, respectively:

$$m_2 \approx \left(\phi^{''}(\mu_{LC})\right)^{-1}$$

$$m_3 \approx -\left(\phi^{''}(\mu_{LC})\right)^{-1}\left(\phi^{'}(\mu_{LC})m_2 + \frac{\phi^{(3)}(\mu_{LC})}{2}m_4\right)$$

$$m_4 \approx 3\left(\phi^{''}(\mu_{LC})\right)^{-1}m_2$$

These relationships are not exhaustive, and one could find many such relationships for even higher orders. The list presented here only concerns results which we will need for our bound on EP.

*Proof.* We will first give an outline of the proof, and then dive into all the equations of the full proof.

The key component of the proof is Stein's lemma (ie: integration by parts). For $LC = \exp\left(-\phi(x)\right)$, it reads: for any statistic $S(x)$ with at-most-polynomial growth:

$$E_{LC}\left(\phi^{'}(x)S(x) - S^{'}(x)\right) = 0 \tag{56}$$

which we will only use for statistics of the form $S_k(x) = (x - \mu_{LC})^k$. This gives us the following relationships:

$$E_{LC}\left(\phi^{'}(x)\right) = 0 \tag{57}$$

$$E_{LC}\left(\phi^{'}(x)(x - \mu_{LC})\right) = 1 \tag{58}$$

$$E_{LC}\left(\phi^{'}(x)(x - \mu_{LC})^2\right) = 0 \tag{59}$$

$$E_{LC}\left(\phi^{'}(x)(x - \mu_{LC})^3\right) = 3m_2 \tag{60}$$

and further relationships of the same form that we won't need. The key intuition in understanding why $LC$ is almost Gaussian is the following: $\phi^{'}(x) \approx \phi^{''}(\mu_{LC})(x - \mu)$. The Stein relationships for $LC$ are thus almost the same relationships that would be obeyed by the Gaussian $g_{\mu_{LC}}(x) = \mathcal{N}\left(x|\mu_{LC}, \left(\phi^{''}(\mu_{LC})\right)^{-1}\right)$. This is why $LC$ is close to $g_{\mu_{LC}}$.

For all these relationships, we will perform a Taylor expansion around $\mu_{LC}$, which now gives us self-consistency relationships between the different moments of $LC$. For example, just keeping the first term in eq. (57) gives us eq. (44):

$$\phi^{'}\left(\mu_{LC}\right) \approx 0$$

We need to be careful with how we deal with the remainder of the Taylor approximation. Using the Taylor-Lagrange formula, we can bound the error that results from cutting off the Taylor series after some term, with a term of the form $C \times (x - \mu)^k$ for some constant C. The expected value under $LC$ of that term can then bounded from the Brascamp-Lieb theorem. For example, to perform the cut-off of eq. (57) we just did, we start from the Taylor-Lagrange expression:

$$\left|\phi^{'}(x) - \phi^{'}(\mu_{LC}) - \phi^{''}(\mu_{LC})(x - \mu_{LC})\right| \leq \frac{K_3}{2}(x - \mu_{LC})^2 \tag{61}$$

which, when we take the expected value, becomes:

$$\left|E_{LC}\left(\phi^{'}(x)\right) - \phi^{'}(\mu_{LC})\right| \leq \frac{K_3}{2}m_2 \leq \frac{K_3}{2\beta_m} \tag{62}$$

where we have applied the Brascamp-Lieb theorem. This concludes the proof of eq. (44), and our introduction to the full proof.

Let's now prove the second relationship of the theorem: eq. (45). We start from eq. (59). We perform the expansion of $\phi^{'}(x)$ up to the $\phi^{''}(\mu_{LC})(x - \mu_{LC})$ term. From Taylor-Lagrange, the error is:

$$\left|\phi^{'}(x) - \phi^{'}(\mu_{LC}) - \phi^{''}(\mu_{LC})(x - \mu_{LC})\right| \leq \frac{K_3}{2}(x - \mu_{LC})^2$$

$$\left|\phi^{'}(x)(x - \mu_{LC})^2 - \phi^{'}(\mu_{LC})(x - \mu_{LC})^2 - \phi^{''}(\mu_{LC})(x - \mu_{LC})^3\right| \leq \frac{K_3}{2}(x - \mu_{LC})^4 \tag{63}$$

We now take the expected value:

$$\left|\phi^{'}(\mu_{LC})m_2 + \phi^{''}(\mu_{LC})m_3 - E_{LC}\left(\phi^{'}(x)\right)\right| \leq \frac{K_3}{2}m_4 \tag{64}$$

Finally, we divide by $m_2$, take out the $\phi^{'}(\mu_{LC})$ term from the absolute value, use the bound on $\frac{m_4}{m_2}$ from eq. (35), and lower bound $\phi^{''}(\mu_{LC})$:

$$
\begin{aligned}
\left|\phi^{''}(\mu_{LC})\frac{m_3}{m_2}\right| &\leq \frac{K_3}{2}\frac{m_4}{m_2} + \left|\phi^{'}(\mu_{LC})\right| \\
&\leq \frac{K_3}{2\beta_m}(3) + \frac{K_3}{2\beta_m} \\
&\leq \frac{2K_3}{\beta_m} \qquad\qquad (65) \\
\left|\frac{m_3}{m_2}\right| &\leq \frac{2K_3}{\beta_m^2} \qquad\qquad (66)
\end{aligned}
$$

which gives us eq. (45).

Now, let's prove the bound on $m_5$ (eq. (46)). The demonstration is quite similar to the $m_3$ bound. We start from another Stein relationship:

$$
E_{LC}\left(\phi^{'}(x)(x-\mu_{LC})^4\right) = 4m_3
$$

With the same Taylor-Lagrange expansion as in eq. (63) and after taking the expected value, we have:

$$
\left|4m_3 - \phi^{'}(\mu_{LC})m_4 - \phi^{''}(\mu_{LC})m_5\right| \leq \frac{K_3}{2}m_6 \qquad\qquad (67)
$$

Which we divide by $m_2$ and manipulate further:

$$
\begin{aligned}
\left|\phi^{''}(\mu_{LC})\frac{m_5}{m_2}\right| &\leq 4\left|\frac{m_3}{m_2}\right| + \left|\phi^{'}(\mu_{LC})\right|\frac{m_4}{m_2} + \frac{K_3}{2}\frac{m_6}{m_2} \\
&\leq \frac{8K_3}{\beta_m^2} + \frac{K_3}{2\beta_m}\frac{3}{\beta_m} + \frac{K_3}{2}\frac{15}{\beta_m^2} \\
&\leq \frac{17K_3}{\beta_m^2} \\
\left|\frac{m_5}{m_2}\right| &\leq \frac{17K_3}{\beta_m^3} \qquad\qquad (68)
\end{aligned}
$$

which gives us eq. (46).

In order to show that any odd centered moment admits a similar bound (as we mention it the main text), we proceed by induction. The Stein relationships:

$$
E\left(\phi^{'}(x)(x-\mu_{LC})^{2k} - 2k(x-\mu_{LC})^{2k-1}\right) = 0
$$

give us the inductive step through steps identical to the preceeding equations, and we have already have the initialization (from eq. 66). We can thus find similar bounds for any higher odd moment of $LC(x)$.

Now we will prove the first order expansions, starting with the one for $m_2$ (eq. (48)). We now start from eq. (58), which is:

$$
E_{LC}\left(\phi^{'}(x)(x-\mu_{LC})\right) = 1
$$

First step, the Taylor-Lagrange expansion. We cut off the Taylor series at $\frac{\phi^{(3)}(\mu_{LC})}{2}(x-\mu_{LC})^2$. We can bound the error with:

$$
\left|\phi^{'}(x)(x-\mu_{LC}) - \phi^{'}(\mu_{LC})(x-\mu_{LC}) - \phi^{''}(\mu_{LC})(x-\mu_{LC})^2 - \frac{\phi^{(3)}(\mu_{LC})}{2}(x-\mu_{LC})^3\right| \leq \frac{K_4}{3!}(x-\mu_{LC})^4 \qquad (69)
$$

which becomes, when we take the expected value:

$$\left| 1 - 0 - \phi^{''}(\mu_{LC})m_2 - \frac{\phi^{(3)}(\mu_{LC})}{2}m_3 \right| \leq \frac{K_4}{3!}m_4$$

$$\left| \phi^{''}(\mu_{LC})m_2 - 1 \right| \leq \frac{1}{2}\left| \phi^{(3)}(\mu_{LC}) \right| |m_3| + \frac{K_4}{3!}m_4$$

$$\left| m_2^{-1} - \phi^{''}(\mu_{LC}) \right| \leq \frac{1}{2}\left| \phi^{(3)}(\mu_{LC}) \right| \left| \frac{m_3}{m_2} \right| + \frac{K_4}{3!}\frac{m_4}{m_2} \tag{70}$$

$$\leq \frac{K_3^2}{\beta_m^2} + \frac{K_4}{2\beta_m} \tag{71}$$

which proves eq. (48), from which eq. (49) is a trivial consequence.

Now, the $m_3$ first order expansion (eq. (50)). We start from the Stein relationship from eq. (59) (which we already used to prove the bound on $\left| \frac{m_3}{m_2} \right|$).

$$E_{LC}\left( \phi^{'}(x)(x - \mu_{LC})^2 \right) = 0$$

The difference between the $m_3$ bound and the $m_3$ first order expansion is that we take a higher-order expansion of $\phi^{'}(x)$. This time, we stop at $\phi^{(4)}(\mu_{LC})(x - \mu_{LC})^3$. The Taylor-Lagrange error is bounded by $\frac{K_5}{4!}(x - \mu_{LC})^4$. This gives us the following bound once we take the expected value.

$$\left| \phi^{'}(\mu_{LC})m_2 + \phi^{''}(\mu_{LC})m_3 + \frac{\phi^{(3)}(\mu_{LC})}{2}m_4 + \frac{\phi^{(4)}(\mu_{LC})}{3!}m_5 \right| \leq \frac{K_5}{4!}m_6 \tag{72}$$

In that equation, $m_5$ is an order of magnitude smaller than the other terms, and we take it out of the absolute value:

$$\left| \phi^{''}(\mu_{LC})m_3 - \left( -\phi'(\mu_{LC})m_2 - \frac{\phi^{(3)}(\mu_{LC})}{2}m_4 \right) \right| \leq \frac{\left| \phi^{(4)}(\mu_{LC}) \right|}{3!}|m_5| + \frac{K_5}{4!}m_6$$

$$\leq \frac{K_4}{3!}\frac{17K_3}{\beta_m^4} + \frac{K_5}{4!}\frac{15}{\beta_m^3}$$

$$\leq \frac{17}{6}\frac{K_3 K_4}{\beta_m^4} + \frac{5}{8}\frac{K_5}{\beta_m^3} \tag{73}$$

which proves eq. (50).

Finally, we prove the last first order expansion: eq. (51) concerning $m_4$. We start from the last Stein relationship: eq. (60):

$$E_{LC}\left( \phi^{'}(x)(x - \mu_{LC})^3 \right) = 3m_2$$

We cut-off the Taylor series after $\frac{\phi^{(3)}(\mu_{LC})}{2}(x - \mu_{LC})^2$. After taking the expected value, the error is:

$$\left| 3m_2 - \phi^{'}(\mu_{LC})m_3 - \phi^{''}(\mu_{LC})m_4 - \frac{\phi^{(3)}(\mu_{LC})}{2}m_5 \right| \leq \frac{K_4}{3!}m_6 \tag{74}$$

In this expression, $\phi^{'}(\mu_{LC})m_3$ and $\frac{\phi^{(3)}(\mu_{LC})}{2}m_5$ are both smaller by an order of magnitude, and we remove them from the absolute value, to finally obtain:

$$\left| \phi^{''}(\mu_{LC})m_4 - 3m_2 \right| \leq \left| \phi^{'}(\mu_{LC}) \right| |m_3| + \left| \frac{\phi^{(3)}(\mu_{LC})}{2} \right| |m_5| + \frac{K_4}{3!}m_6$$

$$\leq \frac{K_3}{2\beta_m}\frac{2K_3}{\beta_m^3} + \frac{K_3}{2}\frac{17K_3}{\beta_m^4} + \frac{K_4}{3!}\frac{15}{\beta_m^3}$$

$$\leq \frac{19}{2}\frac{K_3^2}{\beta_m^4} + \frac{5}{2}\frac{K_4}{\beta_m^3} \tag{75}$$

which proves eq. (51).

In order to find the first order developments of higher order even moments, one proceeds identically to here but from the Stein relationships:

$$E\left(\phi^{'}(x)(x-\mu_{LC})^{2k+1}-(2k+1)(x-\mu_{LC})^{2k}\right) \qquad (76)$$

from which, by the same approach as the proof of eq. 51, we have:

$$m_{2k+2} \approx \frac{(2k+1)}{\phi^{''}(\mu_{LC})}m_{2k} \qquad (77)$$

and by induction, we prove that:

$$m_{2k+2} \approx (2k+1)!!\left[\phi^{''}(\mu_{LC})\right]^{-(k+1)} \qquad (78)$$

which justifies our claim in the main text.

We are only left with proving the final two relationships. For eq. (54), this corresponds to doing a further expansion of the first Stein relationship (eq. (57), from which we proved that $\phi^{'}(\mu_{LC}) \approx 0$):

$$E_{LC}\left(\phi^{'}(x)\right) = 0$$

We stop the Taylor series after $\frac{\phi^{(4)}(\mu_{LC})}{3!}(x-\mu_{LC})^3$. After taking the expected value, we get:

$$\left|\phi^{'}(\mu_{LC})+\frac{\phi^{(3)}(\mu_{LC})}{2}m_2+\frac{\phi^{(4)}(\mu_{LC})}{3!}m_3\right| \le \frac{K_5}{4!}m_4 \qquad (79)$$

We extract the $m_3$ term which is an order of magnitude smaller than the other ones, and obtain:

$$\begin{aligned}
\left|\phi^{'}(\mu_{LC})+\frac{\phi^{(3)}(\mu_{LC})}{2}m_2\right| &\le \left|\frac{\phi^{(4)}(\mu_{LC})}{3!}\right||m_3|+\frac{K_5}{4!}m_4 \\
&\le \frac{K_4}{3!}\frac{2K_3}{\beta_m^3}+\frac{K_5}{4!}\frac{3}{\beta_m^2} \\
&\le \frac{K_3K_4}{3\beta_m^3}+\frac{K_5}{8\beta_m^2} \qquad (80)
\end{aligned}$$

which proves eq. (54).

At last, we reach the proof of eq. (55). We start from the second Stein relationship (eq. (58), which we already used to get the first order expansion of $m_2$):

$$E_{LC}\left(\phi^{'}(x)(x-\mu_{LC})\right) = 1$$

We stop the Taylor series after $\frac{\phi^{(5)}(\mu_{LC})}{4!}(x-\mu_{LC})^4$. After taking the expected value, we get:

$$\left|1-\phi^{''}(\mu_{LC})m_2-\frac{\phi^{(3)}(\mu_{LC})}{2}m_3-\frac{\phi^{(4)}(\mu_{LC})}{3!}m_4-\frac{\phi^{(5)}(\mu_{LC})}{4!}m_5\right| \le \frac{K_6}{5!}m_6 \qquad (81)$$

We divide by $m_2$, then extract the $m_5$ term and obtain:

$$\begin{aligned}
\left|m_2^{-1}-\phi^{''}(\mu_{LC})+\frac{\phi^{(3)}(\mu_{LC})}{2}\frac{m_3}{m_2}+\frac{\phi^{(4)}(\mu_{LC})}{3!}\frac{m_4}{m_2}\right| &\le \left|\frac{\phi^{(5)}(\mu_{LC})}{4!}\right|\left|\frac{m_5}{m_2}\right|+\frac{K_6}{5!}\frac{m_6}{m_2} \\
&\le \frac{K_5}{4!}\frac{17K_3}{\beta_m^3}+\frac{K_6}{5!}\frac{15}{\beta_m^2} \\
&\le \frac{17}{24}\frac{K_3K_5}{\beta_m^3}+\frac{K_6}{8\beta_m^2} \qquad (82)
\end{aligned}$$

proving eq. (55) and concluding our proof. $\qquad \square$

# B   Quality of fixed-points of EP

In this section, we give a detailed proof of our bounds on the quality of the EP approximation.

We assume that all sites $f_i = \exp\left(-\phi_i(x)\right)$ are $\beta_m$-strongly log-concave, with slowly changing log-functions. That is:

$$\forall i, x, \ \phi_i^{''}(x) \ \geq \ \beta_m \tag{83}$$

$$\forall d \in [3, 4, 5, 6] \ \left|\phi_d^{(3)}(x)\right| \ \leq \ K_d \tag{84}$$

The target distribution $p(x)$ then inherits those properties from the sites. Noting $\phi_p(x) = -\log\left(p(x)\right) = \sum_i \phi_i(x)$, then $\phi_p$ is $n\beta_m$-strongly log-concave and for $d \in [3, 4, 5, 6]$,

$$\left|\phi_p^{(d)}(x)\right| \leq nK_d \tag{85}$$

Let $q_i\left(x|r_i, \beta_i\right)$ be the site-approximations of a fixed-point of EP, $q\left(x|r = \sum_i r_i, \beta = \sum_i \beta_i\right)$ be the corresponding approximation of $p(x)$ and $h_i(x)$ the corresponding hybrid distributions. From our hypothesis on the sites, all hybrids are $(\beta_m + \beta_{-i})$-strongly log-concave, with slowly varying log-function (with constants $K_d$). We can thus apply our results from section A to all hybrids and the target distribution.

Some results to keep in mind on the hybrids: first of all,

$$-\frac{\partial \log\left(h_i(x)\right)}{\partial x} = \phi_i^{'}(x) + \beta_{-i}x - r_{-i} \tag{86}$$

This expression is important as it is the one that appears in the Stein relationships.

Also, because $q(x)$ is a Gaussian distribution of mean and variance $\mu_{EP}, v_{EP}$ and with natural parameters $r, \beta$:

$$r \ = \ \beta\mu_{EP} \tag{87}$$

$$\beta \ = \ v_{EP}^{-1} \tag{88}$$

Finally, we have:

$$
\begin{aligned}
\sum_i \beta_{-i}\mu_{EP} &= \sum_{i,j\neq i} \beta_j\mu_{EP} \\
&= (n-1)\sum_j \beta_j\mu_{EP} \\
&= (n-1)\beta\mu_{EP} \\
&= (n-1)r \\
&= (n-1)\sum_j r_j \\
&= \sum_{i,j\neq i} r_j \\
\sum \beta_{-i}\mu_{EP} &= \sum_i r_{-i}
\end{aligned}
\tag{89}
$$

## B.1   Lower-bounding the $\beta_i$

Let's show that we can lower bound the $\beta_i$ at the fixed-point by $\beta_m$.

Recall that $\beta_i$ is obtained from the difference between the inverse variance of $h_i(x)$ and $\beta_{-i}$, and $h_i(x)$ happens to be a $(\beta_m + \beta_{-i})$-strongly log-concave distribution. We can thus apply the

Brascamp-Lieb inequality to the variance:

$$m_2^i \leq \frac{1}{\beta_m + \beta_{-i}} \tag{90}$$

$$\left(m_2^i\right)^{-1} \geq \beta_m + \beta_{-i} \tag{91}$$

Thus, $\beta_i = \left(m_2^i\right)^{-1} - \beta_{-i} \geq \beta_m$ and we have the claimed lower bound.[2]

Thus all hybrids are actually at least $n\beta_m$-strongly log-concave (but could theoretically be stronger. This is one way our bounds can be pessimistic).

### B.2 Approximation of various moments by $q(x)$ and the hybrids

In this section, we will show that some moments of $p(x)$ are matched approximately by the moments of $q(x)$ and/or the moments of the hybrids $h_i(x)$.

We will note $m_k^p$ the $k^{th}$ centered moment of $p(x)$ and $m_k^i$ the moments of the hybrids. We will use $\mu, v$ for the mean and variance of $p(x)$ and $\mu_{EP}, v_{EP}$ for the mean and variance of $q(x)$ and all $h_i(x)$ (recall that, at a fixed-point of EP, $q(x)$ and all $h_i(x)$ share the same mean and variance). The mean and variance have gained special notation due to their special status.

With these notations, the first three even moments of $q$ are respectively $v_{EP}$, $3v_{EP}^2$ and $15v_{EP}^3$, while all odd moments are $0$.

We will show that the following moments are matched:

**Theorem 3.** *When all sites are strongly log-concave with slowly changing log, fixed-points of EP provide a good approximation of several moments of $p(x)$:*

$$\begin{aligned}
\mu &= \mu_{EP} + \mathcal{O}\left(n^{-2}\right) \\
v^{-1} &= v_{EP}^{-1} + \mathcal{O}(1) \\
m_3^p &= \sum_i m_3^i + \mathcal{O}\left(n^{-3}\right) \\
m_4^p &= 3v_{EP}^2 + \mathcal{O}\left(n^{-3}\right) \\
\forall i \; m_4^p &= m_4^i + \mathcal{O}\left(n^{-3}\right)
\end{aligned}$$

*Proof.* Let's first give an outline of the proof.

The logic for all these results is similar. Because all hybrids $h_i(x)$ are $n\beta_m$-strongly log-concave with slowly changing-log, we can apply the results of section A on all those distributions, and obtain inequalities that relate the moments of the $h_i(x)$ to one another. Since they all share the same mean and variance, these become severely constrained. Since $p(x)$ is also log-concave with slowly changing log-function, its mean and variance obey very similar relationships to $\mu_{EP}$ and $v_{EP}$. From the fact that the pair $(\mu, v)$ and the pair $(\mu_{EP}, v_{EP})$ obey almost the same inequalities, we are able to deduce that they are close to one another.

Let's start with $\mu$. From eq. (54), $\mu$ obeys the following simple relationship:

$$\left| \phi_p'(\mu) + \frac{\phi_p^{(3)}(\mu)}{2} v \right| \leq \frac{nK_3 nK_4}{3n^3 \beta_m^3} + \frac{nK_5}{8n^2 \beta_m^2}$$

$$\leq n^{-1}\left( \frac{K_3 K_4}{3\beta_m^3} + \frac{K_5}{8\beta_m^2} \right) \tag{92}$$

Applying the same results to all hybrids $h_i(x)$, we get:

$$\forall i \quad \left| \phi_i^{'}(\mu_{EP}) + \beta_{-i}\mu_{EP} - r_{-i} + \frac{\phi_i^{(3)}(\mu_{EP})}{2}v_{EP} \right| \quad \leq \quad \frac{K_3 K_4}{3n^3\beta_m^3} + \frac{K_5}{8n^2\beta_m^2}$$

$$\leq \quad n^{-3}\frac{K_3 K_4}{3\beta_m^3} + n^{-2}\frac{K_5}{8\beta_m^2} \qquad (93)$$

which is slightly different than eq. (92). Let's now sum the relationship obtained for each $h_i(x)$. The $\beta_{-i}\mu_{EP} - r_{-i}$ terms drop out (eq. (89)) and we get:

$$\left| \phi_p^{'}(\mu_{EP}) + \frac{\phi_p^{(3)}(\mu_{EP})}{2}v_{EP} \right| \leq n^{-2}\frac{K_3 K_4}{3\beta_m^3} + n^{-1}\frac{K_5}{8\beta_m^2} \qquad (94)$$

We have that $\mu$ and $\mu_{EP}$ satisfy almost the same relationship from eq. (92) and (94). We can use this to bound the distance between the two, as a function of the distance between $v$ and $v_{EP}$:

$$\phi_p^{'}(\mu_{EP}) + \frac{\phi_p^{(3)}(\mu_{EP})}{2}v_{EP} = \phi_p^{'}(\mu_{EP}) + \frac{\phi_p^{(3)}(\mu_{EP})}{2}(v + v_{EP} - v)$$

$$\phi_p^{'}(\mu_{EP}) + \frac{\phi_p^{(3)}(\mu_{EP})}{2}v_{EP} - \left( \phi_p^{'}(\mu) + \frac{\phi_p^{(3)}(\mu)}{2}v \right) = \left[ \phi_p^{''}(\xi_1) + \frac{\phi_p^{(4)}(\xi_2)}{2}v \right] (\mu_{EP} - \mu)$$

$$+ \frac{\phi_p^{(3)}(\mu_{EP})}{2}(v_{EP} - v) \qquad (95)$$

where $\xi_1, \xi_2 \in [\mu, \mu_{EP}]$ and we have used first-order expansions at $\mu$ of $\phi_p^{'}(\mu_{EP})$ and $\phi_p^{(3)}(\mu_{EP})$. We can go from upper bounding $\left[ \phi_p^{''}(\xi_1) + \frac{\phi_p^{(4)}(\xi_2)}{2}v \right] (\mu_{EP} - \mu)$ to upper bounding $|\mu - \mu_{EP}|$:

$$\left| \left[ \phi_p^{''}(\xi_1) + \frac{\phi_p^{(4)}(\xi_2)}{2}v \right] (\mu_{EP} - \mu) \right| \geq \min_{\xi_1, \xi_2} \left( \left[ \phi_p^{''}(\xi_1) + \frac{\phi_p^{(4)}(\xi_2)}{2}v \right] \right) |\mu - \mu_{EP}|$$

$$\geq \left[ n\beta_m - \frac{K_4}{2\beta_m} \right] |\mu - \mu_{EP}| \qquad (96)$$

We finally obtain a bound on the distance between $\mu$ and $\mu_{EP}$ by combining eqs. (92), (94), (95) and (96):

$$\left| \left[ \phi_p^{''}(\xi_1) + \frac{\phi_p^{(4)}(\xi_2)}{2}v \right] (\mu_{EP} - \mu) \right| \leq (n^{-1} + n^{-2})\frac{K_3 K_4}{3\beta_m^3} + 2n^{-1}\frac{K_5}{8\beta_m^2} + n\frac{K_3}{2}|v - v_{EP}|$$

$$\leq \mathcal{O}\left(n^{-1}\right) + \mathcal{O}\left(n^{-1}\right) + \mathcal{O}\left(n|v - v_{EP}|\right) \qquad (97)$$

$$|\mu - \mu_{EP}| \leq \mathcal{O}\left(n^{-2}\right) + \mathcal{O}\left(|v - v_{EP}|\right) \qquad (98)$$

Once we show that $v = v_{EP} + \mathcal{O}\left(n^{-2}\right)$, eq. (98) will give us indeed that $\mu = \mu_{EP} + \mathcal{O}\left(n^{-2}\right)$.

Let's now show that $v \approx v_{EP}$. We start from the first order expansion of $m_2^{-1}$ from our extension of the Brascamp-Lieb theorem (eq. (48)). For $p(x)$, this gives us:

$$\left| v^{-1} - \phi_p^{''}(\mu) \right| \leq \frac{n^2 K_3^2}{n^2\beta_m^2} + \frac{nK_4}{2n\beta_m}$$

$$\leq \frac{K_3^2}{\beta_m^2} + \frac{K_4}{2\beta_m} \qquad (99)$$

Again the corresponding relationship for the hybrids is not exactly what we want it to be:

$$\forall i \quad \left| v_{EP}^{-1} - \phi_i^{''}(\mu_{EP}) - \beta_{-i} \right| \leq \frac{K_3^2}{n^2\beta_m^2} + \frac{K_4}{2n\beta_m}$$

$$\leq n^{-2}\frac{K_3^2}{\beta_m^2} + n^{-1}\frac{K_4}{2\beta_m} \qquad (100)$$

But again, we sum all those relationships:

$$\left| n v_{EP}^{-1} - \phi_p^{''}(\mu_{EP}) - (n-1)\beta \right| \leq n^{-1} \frac{K_3^2}{\beta_m^2} + \frac{K_4}{2\beta_m} \tag{101}$$

which further simplifies, because $\beta = v_{EP}^{-1}$, into:

$$\left| v_{EP}^{-1} - \phi_p^{''}(\mu_{EP}) \right| \leq n^{-1} \frac{K_3^2}{\beta_m^2} + \frac{K_4}{2\beta_m} \tag{102}$$

Again, we find that the pairs $(\mu, v)$ and $(\mu_{EP}, v_{EP})$ obey very similar relationships: eqs. (99) and (102). We have:

$$\left| \phi_p^{''}(\mu) - \phi_p^{''}(\mu_{EP}) \right| \leq K_3 \left| \mu - \mu_{EP} \right| \tag{103}$$

and this gives us that $v^{-1} \approx v_{EP}^{-1}$:

$$\left| v^{-1} - v_{EP}^{-1} \right| \leq K_3 \left| \mu - \mu_{EP} \right| + \left( 1 + n^{-1} \right) \frac{K_3^2}{\beta_m^2} + 2 \frac{K_4}{2\beta_m}$$
$$\left| v^{-1} - v_{EP}^{-1} \right| \leq \mathcal{O}(1) + \mathcal{O}\left( n \left| \mu - \mu_{EP} \right| \right) \tag{104}$$

Our final equations for the size of $|\mu - \mu_{EP}|$ and $\left| v^{-1} - v_{EP}^{-1} \right|$ seem to be caught in a loop: you need to know how good one approximation is in order to know how good the second will be and so on. This is not at all the case and it is very easy to cut this loop.

The easiest way is to remark that both $\mu$ and $\mu_{EP}$ are $\mathcal{O}\left( n^{-1} \right)$ away from the mode of $p$ and so they must be $\mathcal{O}\left( n^{-1} \right)$ from one another (see main text, section 2.2). This gives $v^{-1} = v_{EP}^{-1} + \mathcal{O}(1)$ (from eq. (104).

Then, we remark that both $v^{-1}$ and $v_{EP}^{-1}$ are order $n$. The error for $|v - v_{EP}|$ is then of order $n^{-2}$ and we have that $\mu = \mu_{EP} + \mathcal{O}\left( n^{-2} \right)$, from eq. (94). This concludes the first part of our proof.

Let's now look at the fourth moment of the target $m_4^p$. We will show that is matched to by the fourth moment $m_4^i$ of any hybrid and by the fourth moment of the Gaussian approximation of $p(x)$: $3 v_{EP}^2$.

From our Brascamp-Lieb extension, the first order approximation of $m_4^p$ is:

$$\left| \phi^{''}(\mu) m_4^p - 3v \right| \leq n^{-2} \left( \frac{19}{2} \frac{K_3^2}{\beta_m^4} + \frac{5}{2} \frac{K_4}{\beta_m^3} \right) \tag{105}$$

From which, intuitively: $m_4^p \approx 3v \left( \phi_p^{''}(\mu) \right)^{-1} \approx 3v^2 \approx 3 v_{EP}^2$

Let's now formalize this intuition by bounding explicitly each error term:

$$3v^2 - 3 v_{EP}^2 = 6 (v - v_{EP}) \frac{(v + v_{EP})}{2} \tag{106}$$

$$\left| 3v^2 - 3 v_{EP}^2 \right| \leq 6 \left| v - v_{EP} \right| \frac{1}{2 n \beta_m} \tag{107}$$

$$3v \left( \phi_p^{''}(\mu) \right)^{-1} - 3v^2 = 3v \left[ \left( \phi_p^{''}(\mu) \right)^{-1} - v \right] \tag{108}$$

$$\left| 3v \left( \phi_p^{''}(\mu) \right)^{-1} - 3v^2 \right| \leq \left| \left( \phi_p^{''}(\mu) \right)^{-1} - v \right| \frac{3}{n \beta_m} \tag{109}$$

Which we can bound using preceding relationships (eq. (104) and eq. (49)), and which gives us the final bound:

$$\left| m_4^p - 3 v_{EP}^2 \right| \leq n^{-3} \left( \frac{19}{2} \frac{K_3^2}{\beta_m^5} + \frac{5}{2} \frac{K_4}{\beta_m^4} \right) + \frac{6}{n \beta_m} \left| v - v_{EP} \right| + \frac{3}{n \beta_m} \frac{1}{n^2 \beta_m^2} \left[ \frac{2 K_3^2}{\beta_m^2} + \frac{K_4}{2 \beta_m} \right]$$
$$\leq \mathcal{O}\left( n^{-3} \right) \tag{110}$$

Let's note that this final approximation isn't any better of any worse, in terms of orders of magnitude, than the original approximation $m_4^p \approx 3v \left( \phi_p^{''}(\mu) \right)^{-1}$.

Another approximation that is of similar quality, in terms of orders of magnitude, is for any hybrid $i$: $m_4^p \approx m_4^i$. Indeed, from 51 (Brascamp-Lieb extension: $m_4$ first order approximation), we have that:

$$\left| \left[ \phi_i^{''}(\mu_{EP}) + \beta_{-i} \right] m_4^i - 3v_{EP} \right| \leq n^{-4} \frac{19}{2} \frac{K_3^2}{\beta_m^4} + n^{-3} \frac{5}{2} \frac{K_4}{\beta_m^3} \tag{111}$$

and see that $m_4^i$ would obey a similar relationship to $m_4^p$ (eq. (105)) if $\beta_{-i} \approx \sum_{j \neq i} \phi_j^{''}(\mu_{EP})$. That happens to be the case because we also have:

$$\left| v_{EP}^{-1} - \left[ \phi_i^{''}(\mu_{EP}) + \beta_{-i} \right] \right| \leq n^{-2} \frac{2K_3^2}{\beta_m^2} + n^{-1} \frac{K_4}{2\beta_m} \tag{112}$$

Thus, $\phi_i^{''}(\mu_{EP}) + \beta_{-i}$ is approximately constant (in $i$), and approximately equal to $v_{EP}^{-1}$, which is an important result in its own right. If we combine eqs. (111) and (112), we thus have:

$$
\begin{aligned}
m_4^i &= 3v_{EP} \left[ \phi_i^{''}(\mu_{EP}) + \beta_{-i} \right]^{-1} + \mathcal{O}\left( n^{-4} \right) \\
&= 3v_{EP}^2 + \mathcal{O}\left( n^{-4} \right) \\
&= m_4^p + \mathcal{O}\left( n^{-3} \right)
\end{aligned}
\tag{113}
$$

which concludes our proof that all fourth moments of the hybrids and $q$ and $p$ are approximately equal. Note that an absolute error of order $n^{-3}$ translates into a relative error of order $n^{-1}$.

Let's now show how to approximate the third moment of the target $m_3^p$ from the third moments of the hybrids $m_3^i$. We start for the first-order approximation of $m_3^p$ (Brascamp-Lieb extension, eq. (50)):

$$\left| \phi_p^{''}(\mu)m_3^p + \left( \phi^{'}(\mu)v + \frac{\phi^{(3)}(\mu)}{2}m_4^p \right) \right| \leq n^{-2} \left( \frac{17}{6} \frac{K_3 K_4}{\beta_m^4} + \frac{5}{8} \frac{K_5}{\beta_m^3} \right) \tag{114}$$

$$m_3^p \approx -\left( \phi_p^{''}(\mu) \right)^{-1} \left( \phi^{'}(\mu)v + \frac{\phi^{(3)}(\mu)}{2}m_4^p \right) \tag{115}$$

For the hybrids, we have:

$$\forall i \left| \left( \phi_i^{''}(\mu_{EP}) + \beta_{-i} \right) m_3^i + \left( \left( \phi_i^{'}(\mu_{EP}) + \beta_{-i}\mu_{EP} - r_{-i} \right) v_{EP} + \frac{\phi_i^{(3)}(\mu_{EP})}{2}m_4^i \right) \right| \leq n^{-4} \frac{17}{6} \frac{K_3 K_4}{\beta_m^4} + n^{-3} \frac{5}{8} \frac{K_5}{\beta_m^3} \tag{116}$$

We will perform the following steps:

$$m_3^i \approx \left( \phi_i^{''}(\mu_{EP}) + \beta_{-i} \right)^{-1} \left( \left( \phi_i^{'}(\mu_{EP}) + \beta_{-i}\mu_{EP} - r_{-i} \right) v_{EP} + \frac{\phi_i^{(3)}(\mu_{EP})}{2}m_4^i \right) \tag{117}$$

$$\approx v_{EP} \left( \left( \phi_i^{'}(\mu_{EP}) + \beta_{-i}\mu_{EP} - r_{-i} \right) v_{EP} + \frac{\phi_i^{(3)}(\mu_{EP})}{2}m_4^p \right) \tag{118}$$

From which:

$$\sum_i m_3^i \approx v_{EP} \sum \left( \left( \phi_i^{'}(\mu_{EP}) + \beta_{-i}\mu_{EP} - r_{-i} \right) v_{EP} + \frac{\phi_i^{(3)}(\mu_{EP})}{2}m_4^p \right) \tag{119}$$

$$\approx v_{EP} \left( \left( \phi_p^{'}(\mu_{EP}) + 0 \right) v_{EP} + \frac{\phi_p^{(3)}(\mu_{EP})}{2}m_4^p \right) \tag{120}$$

from which we see that $m_3^p$ and $\sum_i m_3^i$ obey very similar relationships (eq. (114) and eq. (120)), and can conclude that they are close.

More formally, starting from eq. (116), let's replace $\phi_i^{''}(\mu_{EP}) + \beta_{-i}$ with $\phi_p^{''}(\mu_{EP})$:

$$
\begin{aligned}
m_3^i &= \left(\phi_i^{''}(\mu_{EP}) + \beta_{-i}\right)^{-1} \left(\left(\phi_i^{'}(\mu_{EP}) + \beta_{-i}\mu_{EP} - r_{-i}\right)v_{EP} + \frac{\phi_i^{(3)}(\mu_{EP})}{2}m_4^i\right) + \mathcal{O}\left(n^{-4}\right) \\
&= v_{EP}\left(\left(\phi_i^{'}(\mu_{EP}) + \beta_{-i}\mu_{EP} - r_{-i}\right)v_{EP} + \frac{\phi_i^{(3)}(\mu_{EP})}{2}m_4^i\right) + \mathcal{O}\left(n^{-2}n^{-2}\right) + \mathcal{O}\left(n^{-4}\right) \quad (121)
\end{aligned}
$$

Now, we replace $m_4^i$ with $m_4^p$. Since, $m_4^i = m_4^p + \mathcal{O}\left(n^{-3}\right)$, we have:

$$
m_3^i = v_{EP}\left(\left(\phi_i^{'}(\mu_{EP}) + \beta_{-i}\mu_{EP} - r_{-i}\right)v_{EP} + \frac{\phi_i^{(3)}(\mu_{EP})}{2}m_4^p\right) + \mathcal{O}\left(n^{-4}\right) \quad (122)
$$

which we finally sum for $i$: the $\beta_{-i}\mu_{EP} - r_{-i}$ sum to 0, leaving:

$$
\sum_i m_3^i = v_{EP}\left(\phi_p^{'}(\mu_{EP})v_{EP} + \frac{\phi_p^{(3)}(\mu_{EP})}{2}m_4^p\right) + \mathcal{O}\left(n^{-3}\right) \quad (123)
$$

Because, $\mu = \mu_{EP} + \mathcal{O}\left(n^{-2}\right)$ and $v = \left(\phi_p^{''}(\mu)\right)^{-1} + \mathcal{O}\left(n^{-2}\right) = v_{EP} + \mathcal{O}\left(n^{-2}\right)$, $\sum_i m_3^i$ and $m_3^p$ have identical first order expansions (which is of order $n^{-2}$). More precisely:

$$
\begin{aligned}
\phi_p^{'}(\mu_{EP})v_{EP} + \frac{\phi_p^{(3)}(\mu_{EP})}{2}m_4^p &= \phi_p^{'}(\mu_{EP})v + \frac{\phi_p^{(3)}(\mu_{EP})}{2}m_4^p + \mathcal{O}\left(n^{-2}\right) \quad (124) \\
&= \phi^{'}(\mu)v + \frac{\phi^{(3)}(\mu)}{2}m_4^p + \mathcal{O}\left(n^{-2}\right) + \mathcal{O}\left(|\mu - \mu_{EP}|\right) \quad (125)
\end{aligned}
$$

because: $\left|\phi^{'}(\mu) - \phi^{'}(\mu_{EP})\right| \leq n\beta_m\left|\mu - \mu_{EP}\right|$ and, similarly, $\phi^{(3)}(\mu) - \phi^{(3)}(\mu_{EP}) = \mathcal{O}\left(n\left|\mu - \mu_{EP}\right|\right)$. And:

$$
\begin{aligned}
(v - v_{EP})\left(\phi_p^{'}(\mu_{EP})v_{EP} + \frac{\phi_p^{(3)}(\mu_{EP})}{2}m_4^p\right) &= \mathcal{O}\left(|v - v_{EP}|\right)\left(\mathcal{O}\left(1\right)\mathcal{O}\left(n^{-1}\right) + \mathcal{O}\left(n\right)\mathcal{O}\left(n^{-2}\right)\right) \\
&= \mathcal{O}\left(n^{-3}\right) \quad (126)
\end{aligned}
$$

Which gives us the final expression:

$$
m_3 = \sum_i m_3^i + \mathcal{O}\left(n^{-3}\right) \quad (127)
$$

which concludes our proofs on the quality of the EP approximation.

In the main text, we have also used the following relationship, detailing the second order expansion of $v_{EP}^{-1}$:

$$
v_{EP}^{-1} = \phi_p^{''}(\mu_{EP}) + \sum_i\left[\phi_i^{(3)}(\mu_{EP})\frac{m_3^i}{2v_{EP}}\right] + \phi_p^{(4)}(\mu_{EP})\frac{3v_{EP}^2}{3!v_{EP}} + \mathcal{O}\left(n^{-1}\right) \quad (128)
$$

For the inquisitive reader, this is obtained by starting from our Brascamp-Lieb extension, eq. (55), applied to all hybrids. Then proceeding to approximate $m_4^i \approx 3v_{EP}^2$ and summing. $\qquad\square$

## Footnotes

[2]By the the same logic, if all sites are strongly log-concave, the dynamics of EP must always maintain $\beta_i \geq \beta_m$. It is thus useless to initialize the EP algorithm at a lower value.