[Reviews · NeurIPS 2015]

Submitted by Assigned_Reviewer_1

This paper describes some bounds for the error made in the estimation of the posterior mean and variance of a target distribution when using expectation propagation. The bounds obtained compare well with the corresponding bounds of the Laplace approximation, which motivates the use of EP over such a method. A practical limitation of the paper is that it does not contain experiments assessing the results obtained.
Summary: Interesting theoretical results about expectation propagation. The lack of experiments questions the quality of the paper.

Submitted by Assigned_Reviewer_2

1. Given that the main goal is to estimate the mean, the mode approximation seems to be not the right comparison. I would suggest to include some comparison with other methods that directly estimate mean, including Monte Carlo (which should be O(1/(n\sqrt{m})), where $m$ is the sample size, here the $1/n$ factor is due to the concentration of p(x) itself), and quadrature based methods (O(1/(nm))), which should match with EP with m = n.

See, e.g., Huszar & Duvenaud Optimally weighted herding is Bayesian quadrature.

It is also worth to mention the asymptotic analysis of the other variational inference methods. For example, see Wang, Bo, and D. M. Titterington. "Convergence and asymptotic normality of variational Bayesian approximations for exponential family models with missing values." Proceedings of the 20th conference on Uncertainty in artificial intelligence. AUAI Press, 2004.

2. All the $f_i(x)$ are assumed to be $\beta_m$ log-concave. How would the result change when $\beta_m$ are different for different $f_i(x)$?

typos:

line 249: "mean x^*_{LC}" appendix line 912: "\beta_i is obtained the difference" appendix line 914: "to to"
Summary: This paper establishes theoretical analysis of EP

for log concave distributions (with slow changing) under the asymptotic setting when the number n of "sites" (corresponding data points) increases to infinite. The main proof technique is based on an extension of Brascamp-Lieb inequality which is of independent interest.

This work can have a potential significant importance given that theoretical properties of EP have not been well discussed before.

Submitted by Assigned_Reviewer_3

The paper derives inference bounds for the Expectation-Propagation (EP) algorithm in variational inference. The results are developed for log-concave distributions with slowly-changing logs, relating the finite-sample bounds to these obtained via Bernstein-von Mises limits for the Canonical Gaussian Approximation (CGA). Kullback-Leibler divergence costs of EP versus CGA are also discussed.

EP is a popular method that works well in practice but comes with few theoretical guarantees, so these developments are of interest.

There are of course several directions in which these results could be improved, which the authors already discuss. In practice, the current results are still restrictive, but they are certainly a step in the right direction.

My main discussion point is that EP is often more powerful for prediction than inference, but the paper only discusses inference approximation bounds. I expect that, even in cases when the sites are not log-concave, EP could perform badly in inference but reasonably in prediction. In that sense, the current bounds are not as interesting as they could be.

Some minor points:

* the notation $\phi_p$ to denote $\sum \phi_i$ is a bit confusing. * many of the formulae don't have punctuation marks at the end.

* The authors could cite "Expectation propagation as a way of life" by Andrew Gelman, Aki Vehtari, Pasi Jylaenki, Christian Robert, Nicolas Chopin, John P. Cunningham. * The references should be cleaned up.

Summary: The paper is nicely written and it provides some theoretical guarantees for Expectation-Propagation which were previously unavailable. EP is widely used so these results should be of interest.

Submitted by Assigned_Reviewer_4

This paper presents a novel approach to the analysis of expectation propagation (EP), proving several new results about its accuracy.

The paper is well-written and the methods provide a good foundation for future work in this area.

The main limitation, admitted by the authors, is that the theorems have overly restrictive conditions that rarely hold in practice.

So the paper's results do not directly apply to common uses of EP.

I was able to follow the derivations fairly easily except for equation (14).

In (14), beta_(-i) and r_(-i) are used without definition.

To make (14) more intuitive, it would help to add one sentence mentioning that hybrids are n*beta_m strongly log-concave.

The discussion of fixed points in section 1.4 is a bit odd since there are known cases where EP does not have fixed points, and sufficient conditions for a fixed point to exist were already given by Minka (UAI 2001).

These conditions are that factors must be bounded and the moments must exist.

If the factors are strongly log-concave, as in this paper, then the conditions are satisfied so a fixed point exists.

Obviously, if moments don't exist (e.g. a Cauchy distribution) then EP cannot have a fixed point.

Summary: Novel theory about expectation propagation that is not quite strong enough to apply to practice, but a good starting point.

Author Feedback
Author rebuttal: Response to reviewers

Dear reviewers,

Thank you all very much for your time and your comments on how to improve our presentation of our results.

A common theme from your comments is a concern on whether our results and our methods are of general interest.

We agree that the assumptions we use are restrictive. However, the proof technique we introduce is flexible and will extend to a more general setup (at the cost of heavier notation and longer calculations). The easiest extension is to suppose only that hybrid distributions are strongly log-concave, which is much less stringent than making assumptions on likelihood sites, and covers many latent Gaussian models, like logistic or probit regression. We have already managed to extend Theorem 1 to this case, and we therefore believe our proofs could serve as blueprints for more general results. Note that our results are non-probabilistic and make no assumptions on the data-generating process Most results of good behavior of statistical algorithms are of the form "probability of bad behavior goes to 0", whereas we give results showing that it's impossible for EP not to work under our assumptions, under purely functional assumptions on the sites If one were willing to make assumptions on the data-generating process, one could also be less stringent on what the sites should be like, giving a second avenue for extending our results. Finally, note that, as pointed out by referee 4, EP has proven time and time again to be extremely resilient to theoretical analysis in the 15 years since it has been proposed. Our results, while certainly restrictive and limited, represent a sizable step forward from the current state of the art.

Many of you also feel that the absence of simulations is a weakness of the paper. We chose to focus on theoretical results because the literature already contains many simulations studies comparing EP to other methods. Since the practical effectiveness of EP is not in doubt, we tried to explain as much of our results as the space constraints would allow.

Thank you again for your time in reviewing this work.

Best regards ,

The authors

Responses to specific comments

Reviewer 1: The CGA was meant as a baseline approximate inference method. Comparaison against other methods is interesting, but we feel like a more honest comparaison would also include computational cost of each method, and goes way beyond the scope of our work. It would also overlap with work such as Nickisch and Rasmussen, 2008. Having multiple beta_m surprisingly changes nothing. Indeed, in that case all hybrids and the priors are still (\sum beta_m)-strongly log-concave.

Reviewer 2: Using our results to study prediction performance is a great idea. It seems to us that Th. 1 could be used to bound prediction error, relative to perfect Bayesian inference, in a classification setting where predictions are made based on posterior means and variances (using smoothness in the prediction function). . Also, while we show that EP performs well on log-concave sites, we believe that it could also work well on non log-concave sites, both for prediction and classification.

Reviewer 3: Thank you very much for correcting our mistake about the existence of fixed-points. We'll be sure to modify this point in future presentation.

Reviewer 6: Our results are indeed specific to Gaussian EP. However, Gaussian EP does represent an overwhelming majority of the practical uses of EP. It thus seems to us like this isn't very damaging for the generality of our results. It seems very hard to extend our current methods to non-Gaussian approximating families.